# Molecular parallelisms between pigmentation in the avian iris and the integument of ectothermic vertebrates

Pedro Andrade[1☯‡]*, Małgorzata A. Gazda[2☯‡], Pedro M. Araújo[1,3☯‡], Sandra Afonso[1], Jacob. A. Rasmussen[4,5], Cristiana I. Marques[1,6], Ricardo J. Lopes[1], M. Thomas P. Gilbert[4,7,8], Miguel Carneiro[1,6]*

1 CIBIO, Centro de Investigação em Biodiversidade e Recursos Genéticos, InBIO Laboratório Associado, Universidade do Porto, Vairão, Portugal, 2 Institut de Biologie de l'Ecole Normale Supérieure (IBENS), Ecole Normale Supérieure, CNRS, INSERM, Université PSL, Paris, France, 3 MARE–Marine and Environmental Sciences Centre, Department of Life Sciences, University of Coimbra, Coimbra, Portugal, 4 Center for Evolutionary Genomics, Faculty of Science, University of Copenhagen, Copenhagen, Denmark, 5 Laboratory of Genomics and Molecular Medicine, Department of Biology, University of Copenhagen, Copenhagen, Denmark, 6 Departamento de Biologia, Faculdade de Ciências, Universidade do Porto, Porto, Portugal, 7 The GLOBE Institute, Faculty of Health and Biomedical Sciences, University of Copenhagen, Copenhagen, Denmark, 8 University Museum, Norwegian University of Science and Technology, Trondheim, Norway

☯ These authors contributed equally to this work.
‡ These authors share first authorship on this work.
* pandrade@cibio.up.pt (PA); miguel.carneiro@cibio.up.pt (MC)

**Data Availability Statement:** Whole-genome sequencing data and RNA sequencing data are available in the Sequence Read Archive (www.ncbi.nlm.nih.gov/sra) under BioProject PRJNA679326.

## Abstract

Birds exhibit striking variation in eye color that arises from interactions between specialized pigment cells named chromatophores. The types of chromatophores present in the avian iris are lacking from the integument of birds or mammals, but are remarkably similar to those found in the skin of ectothermic vertebrates. To investigate molecular mechanisms associated with eye coloration in birds, we took advantage of a Mendelian mutation found in domestic pigeons that alters the deposition of yellow pterin pigments in the iris. Using a combination of genome-wide association analysis and linkage information in pedigrees, we mapped variation in eye coloration in pigeons to a small genomic region of ~8.5kb. This interval contained a single gene, *SLC2A11B*, which has been previously implicated in skin pigmentation and chromatophore differentiation in fish. Loss of yellow pigmentation is likely caused by a point mutation that introduces a premature STOP codon and leads to lower expression of *SLC2A11B* through nonsense-mediated mRNA decay. There were no substantial changes in overall gene expression profiles between both iris types as well as in genes directly associated with pterin metabolism and/or chromatophore differentiation. Our findings demonstrate that *SLC2A11B* is required for the expression of pterin-based pigmentation in the avian iris. They further highlight common molecular mechanisms underlying the production of coloration in the iris of birds and skin of ectothermic vertebrates.

https://www.ncbi.nlm.nih.gov/bioproject/?term=
PRJNA679326.

**Funding:** This work was supported by the
Fundação para a Ciência e Tecnologia (FCT, https://
www.fct.pt) through POPH-QREN funds from the
European Social Fund and Portuguese MCTES
(CEECINST/00014/2018/CP1512/CT0002); by
Portuguese national funds to R.J.L. (Transitory
Norm contract [DL57/2016/CP1440/CT0006]); by
research fellowships attributed to M.A.G. (PD/BD/
114042/2015) and C.I.M. (SFRH/BD/147030/2019)
in the scope of the Biodiversity, Genetics, and
Evolution (BIODIV) PhD program; by ERC
Consolidator Grant number 681396 to M.T.P.G.
(https://erc.europa.eu); by an EMBO Short Term
Fellowship number 8033 to M.A.G. (https://www.
embo.org); by the project "PTDC/BIA-EVL/31569/
2017 - NORTE -01-0145-FEDER-30288", co-
funded by NORTE2020 through Portugal 2020 and
FEDER Funds, and by National Funds through FCT
and by National Funds through FCT in the scope of
the project UIDB/50027/2020 (Base). The funders
had no role in study design, data collection and
analysis, decision to publish, or preparation of the
manuscript.

**Competing interests:** The authors have declared
that no competing interests exist.

## Author summary

Eye color is an important component of ornamental diversity in birds, resulting from the
interactions between pigments and scattering elements in specialized cells in the iris.
These cells share many structural and chemical characteristics with pigment cells found in
the dermis of fish, amphibians and reptiles. In this study, we took advantage of variation
in eye color found in domestic pigeons, which can be either pigmented (wild-type, due to
deposition of pterins) or unpigmented (pearl-eye). Using a combination of genomic and
transcriptomic analyses, we show that the ability to express pterin pigmentation is
explained by *SLC2A11B*, a gene that has been previously implicated in the differentiation
of pigment cells in the skin of fish. Our results together with cellular and pigmentary
observations support an evolutionary and developmental link between the iris of birds
and the skin of ectotherms.

## Introduction

Coloration plays a vital role in the life history of many animals, so understanding the cellular
and molecular underpinnings of traits related to ornamentation, camouflage or aposematic
signals is key to shed light on major questions in evolutionary research [1]. In many species of
vertebrates, colorful ornaments are linked to the development of specialized dermal cell types
termed chromatophores, typically organized in a so-called "dermal chromatophore unit"
whose basic structure is largely conserved in fish, amphibians and reptiles [2]. With few excep-
tions, this unit is formed by several types of chromatophores, including pterin and carotenoid-
containing xanthophores, iridophores with reflecting guanine platelets, and melanophores
containing melanin pigments. This arrangement is capable of generating a stunning variety of
colorful ornaments, but is not found in the integument of endothermic vertebrates. In endo-
therms, it is likely that the evolution of insulating epidermal integument, such as hair and
feathers (in which pigments are deposited) has led to the loss of function of dermal pigments
and structures that generate color [3].

A less well-studied source of ornamental diversity is the eye. Eye color arises from the depo-
sition of pigments in the pigmented epithelium of the iris [4], and is thought to play many of
the same signaling functions of integumentary ornaments, both in intra-specific and inter-spe-
cific communication [5–9]. Dermal chromatophores and the iris pigmented epithelium share
a common developmental origin from neural crest cells [10–11], and similarities in ultrastruc-
ture and pigment type composition have been described [3,12]. In birds, iris pigmentation is
frequently associated with the deposition of pterins [13], a class of pigments that is common in
the skin of ectothermic vertebrates, but virtually absent in the integument of birds and mam-
mals. This observation together with the presence of all dermal chromatophore types in the
avian iris–similar to those found in the skin of fish, amphibians and reptiles–have led previous
authors to propose the iris as an "evolutionary refugium" for pigment cells in endothermic ver-
tebrates [3]. Investigating genes associated with eye color variation in birds could shed light on
this hypothesis, but finding suitable biological models is difficult given that differences in eye
color are generally fixed between species [14].

A good model to investigate the molecular determinants of eye color variation and pigment
cell evolution in birds is the domestic pigeon (*Columba livia*, Fig 1). Pigeons typically have
eyes exhibiting a range of yellow and red hues, which arise from the deposition of pterin pig-
ments in the anterior surface of the iris combined with strong vascularization [15]. In addition
to the wild-type eye color phenotype, mutant phenotypes such as the pearl-eye arose during

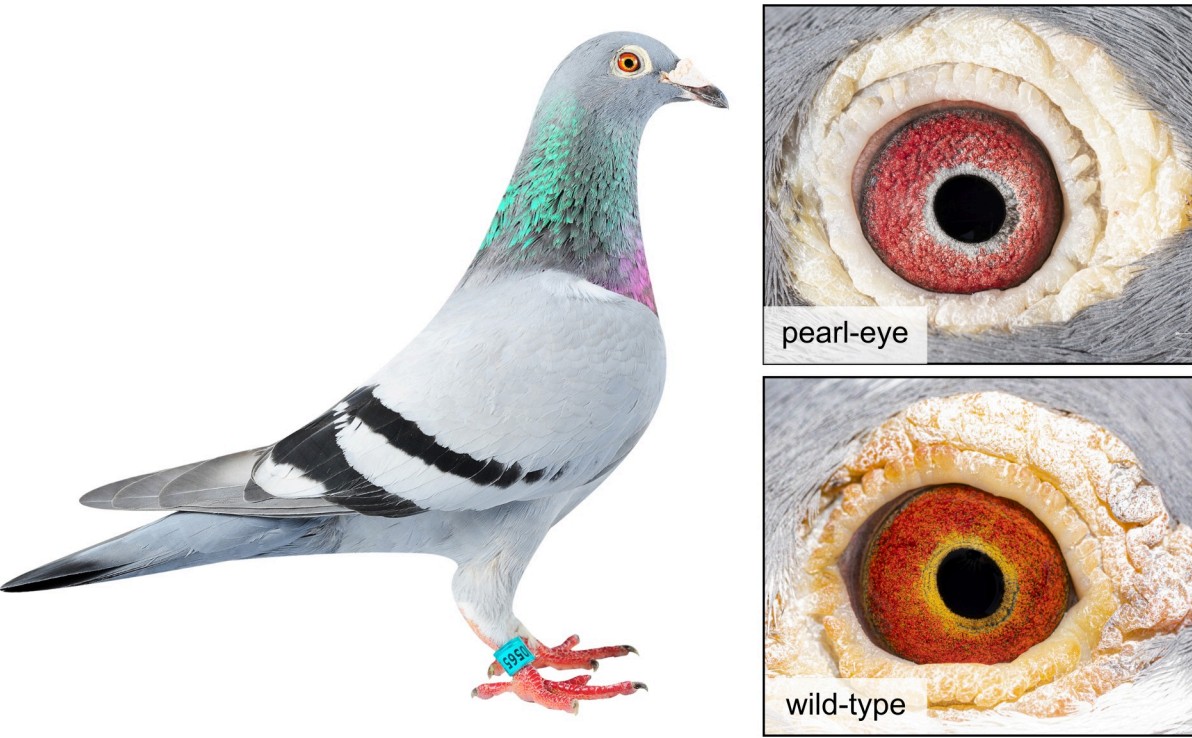

**Fig 1. Eye color in domestic pigeons (*Columba livia*).** Wild-type individuals exhibit a pigmented yellow iris, while pearl-eye mutants (controlled by a recessive allele) show unpigmented eyes. The yellow coloration of wild-type pigeons is due to the deposition of pterins in the pigmented epithelium of the iris. Photo credits: P. M. Araújo.

domestication [16]. Electron microscopy imaging shows that pigment cells in the iris of pearl-eye pigeons are structurally identical to pigment cells in wild-type birds, but lack yellow pterin pigments (the red coloration in both eye color types is due to blood vessels) [15]. Classical genetic studies indicate an autosomal recessive mode of inheritance for pearl-eye [16], as presumably birds carrying two copies of the mutant allele (*tr/tr*) are impaired in their capacity to synthesize or deposit pterins in the iris compared to wild-type birds (*Tr/Tr* or *Tr/tr*).

In this study, we used a combination of genetic, genomic and gene expression analyses to reveal the molecular mechanism underlying eye color variation in the domestic pigeon. Our findings demonstrate that pearl-eye maps to a small 8.5 kb autosomal segment containing a single gene, solute carrier family 2 member 11b (*SLC2A11B*). A nonsense mutation occurring within this interval is the most likely causal variant for the phenotype. As this gene had been previously implicated in chromatophore differentiation in fish, our results support the hypothesis that the avian iris shares an evolutionary and developmental link with dermal chromatophores found in the integument of ectothermic vertebrates.

## Results

### Genetic mapping of the genomic region explaining the pearl-eye phenotype

To investigate the genetic basis of the pearl-eye phenotype in domestic pigeons, we carried out whole genome sequencing of 29 racing pigeons exhibiting wild-type (n = 14) and pearl-eye pigmentation (n = 15). These data were combined with published whole-genome sequencing data of 20 individuals [17] derived from breeds that are expected to be fixed for one type of eye-color. In total, the genome-wide analysis described below was conducted in a total of 26

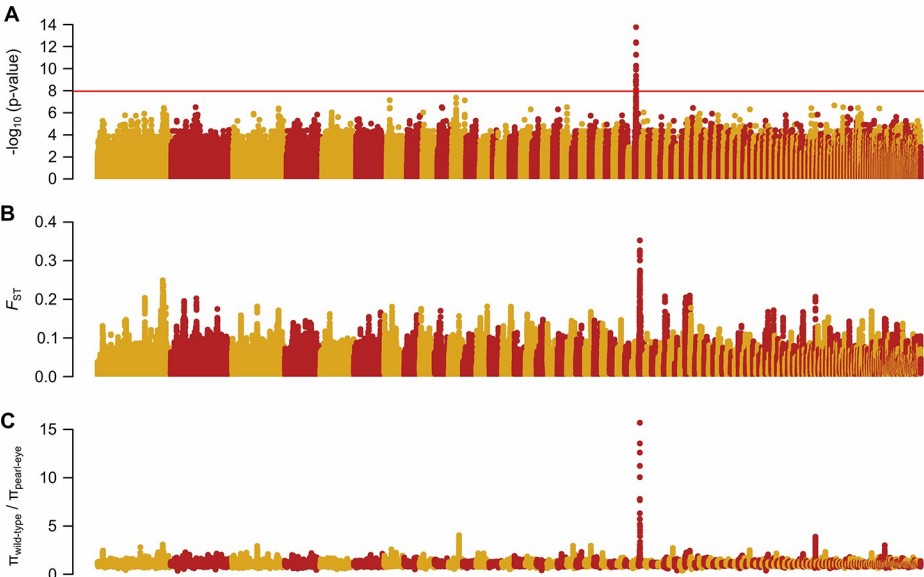

**Fig 2. Genetic mapping of the pearl-eye locus in domestic pigeons.** (A) $-\log_{10}$ transformation of Fisher's exact test
$P$-values under a recessive model of association. Each dot represents an individual SNP. The red solid line indicates the
Bonferroni-corrected critical value ($P = 1.13\text{x}10^{-8}$). (B) $F_{ST}$ scan across the genome between wild-type and pearl-eye
pigeons. Each dot represents $F_{ST}$ averaged over 20 kb windows and iterated in steps of 4 kb across each scaffold. (C)
Ratio of nucleotide diversity ($\pi$) in wild-type and pearl-eye pigeons. Each dot represents the ratio averaged over 20 Kb
windows and iterated in steps of 4 kb across each scaffold. In all three panels, the different scaffolds are presented on
the x-axis in the same order as they appear in the rock pigeon reference genome assembly.

wild-type and 23 pearl-eye birds (S1 Table). The sequencing reads were mapped against the
rock pigeon reference genome [18], resulting in an average coverage per individual of 11.5X
(range 5.2–35.4; S1 Table).

We compared the genomes of wild-type and pearl-eye pigeons using several statistics that
utilize different aspects of the data. First, we screened the genome for variants associated with
pearl-eye pigmentation by testing for significant differences between the distributions of geno-
types in wild-type versus pearl-eye pigeons under a recessive Mendelian model. A single region
of the rock pigeon reference genome reached genome-wide significant levels after Bonferroni
correction (Fisher's exact test, $P < 1.13\text{x}10^{-8}$; 4,411,065 tests; Fig 2A). This region contained 59
significant SNPs defining a stretch of 459 kb on scaffold 30 (AKCR02000030.1:1,494,616–
1,929,542bp). Second, we used two statistics summarizing genetic differentiation and diversity
across the genome between wild-type and pearl-eye pigeons: 1) the fixation index ($F_{ST}$); and 2)
the ratio of nucleotide diversity in wild-type pigeons and pearl-eye pigeons ($\pi_{\text{wild-type}}/\pi_{\text{pearl-}}$
$_{\text{eye}}$)–the pearl-eye phenotype has risen in frequency during domestication, and therefore, low
genetic diversity around the causative locus is expected. Both statistics were calculated in win-
dows of 20kb iterated every 4kb across each scaffold. The same region on scaffold 30 contained
the top values of the empirical distributions of both statistics ($F_{ST}$–top 18 windows; $\pi_{\text{wild-type}}/$
$\pi_{\text{pearl-eye}}$–top 14 windows; Fig 2B and 2C).

To confirm the association between our candidate region on scaffold 30 and the pearl-eye
phenotype, we further obtained phenotype and genotype data in pedigrees. We screened 26
parent-offspring trios where both phenotypes were present, and genotyped a variant with diag-
nostic alleles between wild-type and pearl-eye pigeons chosen from the whole-genome
sequencing data that was also informative between parents in the 26 trios. We found that all 20
wild-type birds and 6 pearl-eye birds followed the expected inheritance pattern. Two

additional regions, on scaffolds 7 and 11, evidenced a moderate (non-significant) signal of association in our genomic scans, but individual genotypes in both regions did not follow the expected inheritance pattern. Overall, the combination of whole-genome sequencing and family-based analyses strongly suggests that the interval on scaffold 30 contains the gene associated with the pearl-eye phenotype.

## An 8.5 kb haplotype containing a single gene is associated with pearl-eye

To increase resolution within the candidate region, we used identical-by-descent (IBD) mapping. The pearl-eye phenotype has likely emerged a single time during domestication, and therefore, the causal mutation should be located within an interval where all pearl-eye individuals are homozygous for a common haplotype. In agreement with this expectation, we found that all pearl-eye birds were nearly devoid of polymorphism for a ~20 kb interval (AKCR02000030.1:1,880,950–1,902,260bp; Fig 3A). Importantly, two individuals with wild-type phenotype were homozygous for the same haplotype at the 5'end of the 20kb interval, allowing us to reduce the size of the candidate region to ~8.5kb (AKCR02000030.1:1,893,852–1,902,260bp; Fig 3A). An examination of the annotation of the rock pigeon reference genome revealed that a single gene was located in this genomic interval: solute carrier family 2 member 11b (*SLC2A11B*). This gene is an excellent candidate for mediating iris pigmentation in pigeons since it has been previously shown to be required for xanthophore differentiation in medaka fish [19].

## A nonsense mutation in *SLC2A11B* segregates with iris pigmentation

We next screened the 8.5kb candidate interval for potential causative mutations, including single-base variants and larger structural changes such as inversions, copy number variation, and indels. A single point mutation with a potential impact on protein function of *SLC2A11B* stood out (AKCR02000030.1:1,895,934bp). This mutation creates a premature STOP codon in exon 3 of the likely canonical *SLC2A11B* transcript and overlaps a position that shows strong evolutionary conservation in whole-genome alignments of multiple bird species (Figs 3B and S1). We Sanger sequenced this SNP for the set of samples used for whole-genome sequencing (n = 29) and an additional cohort of 63 samples: all 40 pearl-eye birds were homozygous for the nonsense allele, whereas all 52 wild-type birds were either homozygous for the wild-type allele or heterozygous. The nonsense mutation follows the expected inheritance pattern and is thus a good candidate to underlie the pearl-eye phenotype.

The candidate nonsense mutation in *SLC2A11B* is likely to lead to an accumulation of aberrant gene products. To test this, we carried out *de novo* transcriptome assemblies of one individual of each phenotype using RNA-sequencing (RNA-seq) data and compared the inferred transcripts to those available from the chicken and zebra finch genome annotation. In line with our expectations, the pearl-eye transcriptome contained highly truncated *SLC2A11B* transcripts that did not match the likely canonical transcript recovered from the wild-type individual (S2 Fig).

## Low expression of *SLC2A11B* in pearl-eye pigeons

The presence of a nonsense mutation in *SLC2A11B* in pearl-eye pigeons raises the possibility that it might be the target of cellular mechanisms that promote transcript degradation. We generated RNA-seq data of four wild-type and two pearl-eye birds (S2 Table) and profiled gene expression for 13,615 genes. Our differential expression analysis revealed only a modest number of differentially expressed genes (n = 64) with 24 genes up-regulated in pearl-eye individuals and 40 down-regulated, Fig 4A, S3 Table). Importantly, *SLC2A11B* was one of the

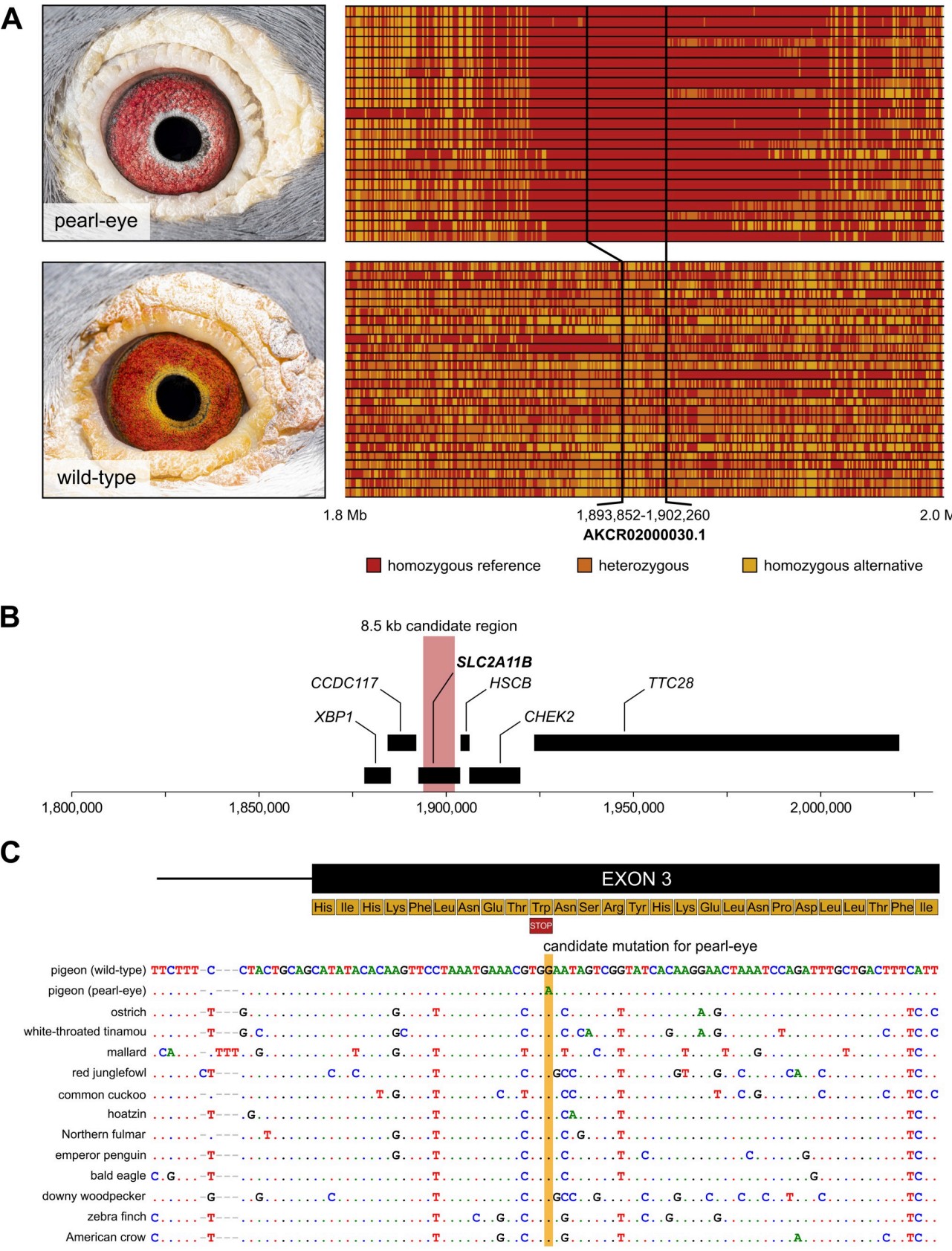

**A**

pearl-eye

wild-type

1.8 Mb          1,893,852-1,902,260          2.0 Mb
**AKCR02000030.1**

■ homozygous reference   ■ heterozygous   ■ homozygous alternative

**B**

8.5 kb candidate region

*SLC2A11B*

*CCDC117*          *HSCB*          *TTC28*

*XBP1*          *CHEK2*

1,800,000          1,850,000          1,900,000          1,950,000          2,000,000

**C**

EXON 3

His Ile His Lys Phe Leu Asn Glu Thr Trp Asn Ser Arg Tyr His Lys Glu Leu Asn Pro Asp Leu Leu Thr Phe Ile
STOP

candidate mutation for pearl-eye

pigeon (wild-type)          TTCTTT-C---CTACTGCAGCATATACACAAGTTCCTAAATGAAACGTGGAATAGTCGGTATCACAAGGAACTAAATCCAGATTTGCTGACTTTCATT
pigeon (pearl-eye)          .....-.---.........................................A......................................
ostrich          .....T--G.............G....T......C...C......T.....A.G...............TC.C
white-throated tinamou          .....T--G.C..........GC.........C...C.CA...T...G...A.G.......T........C.TC.C
mallard          .CA...-.TTT..G.......T...G...T........T...T..C.T.....T....T..G..........T.....TC..
red junglefowl          .....CT--.......C..C......T......C..GCC.....T....GT...G..C.....CA..C........TC..
common cuckoo          .....-.---..........T.G..T.....C.T....CC.....T....T..C.G......C.......C.TC.C
hoatzin          .....T--.G.............T......C...CA...T...................TC..
Northern fulmar          .....-.---...T.........G..T......C...C.G...T.................TC..
emperor penguin          .....T--...........G..T......C...C......T.C...........C......G.......TC..
bald eagle          C.G...T--...........T......C...C......T...............G........TC..
downy woodpecker          .....G---..G.......C........T......C..GCC..G....C..G...C..C..C..T...C........TC..
zebra finch          C.....T--...........T....C..G.C...G......T.C.G...G.............TC..
American crow          C.....T--...........T......G..C...G......T..............A.....C..TC..

**Fig 3. Fine mapping of the causal region for the pearl-eye phenotype.** (A) Genotyping across the candidate causal region. Each line corresponds to one individual, and each column to a single-nucleotide polymorphism. Color of each individual cell indicates the genotype, which can be either homozygous reference (red), heterozygous (orange) or homozygous alternative (yellow). We note that the reference genome contains the pearl eye haplotype. Pearl-eye pigeons are homozygous for a stretch of sequence of ~20kb. This interval can be further reduced by excluding a segment in which two wild-type individuals are homozygous, resulting in a ~8.5kb region. (B) Gene content along the candidate region. The 8.5kb interval identified in panel (A), here highlighted in red, contains a single protein coding gene: *SLC2A11B*. (C) Alignment of sequences of multiple avian species around the candidate causal mutation in *SLC2A11B*, indicating strong sequence conservation at the candidate position except for the pearl-eye haplotype. A scheme of the partial amino acid content at the 5' end of exon 3 and the location of the premature STOP codon are shown. Additional bird species are represented in S1 Fig.

differentially expressed genes and found to be down-regulated in pearl-eye pigeons (log fold-change = -1.54; FDR-adjusted $P = 3.76 \times 10^{-4}$). We confirmed the observed differences through reverse transcription quantitative polymerase chain reaction (RT-qPCR) in a larger number of samples (wild-type n = 9; pearl-eye n = 5; Mann-Whitney U = 0, $P = 3.35 \times 10^{-3}$, Fig 4B). Two other genes located in the same linkage group as *SLC2A11B* were also differentially expressed: *TRAFD1* (located ca. 1.6 Mb from the candidate interval) and *CHEK2* (located ca. 4 kb away). While we cannot rule out the possibility of *cis*-regulatory variants within our candidate region affecting the expression of these nearby genes (particularly for *CHEK2*), their known functions do not suggest any direct link to pigment synthesis, transport or deposition that could explain differences in eye color.

Next, we used the RNA-seq data to measure allele-specific expression differences in three wild-type individuals that were heterozygous for both alleles. We quantified the number of reads belonging to each allele at the candidate nonsense mutation. This revealed a significant deviation from the null expectation of equal representation of both alleles, with the pearl-eye allele exhibiting at least a 4-fold reduction in expression in all three individuals when compared to the wild-type allele (Fisher's exact test, $P = 1.55 \times 10^{-12}$; Fig 4C, S2 Table). Overall, our expression studies are consistent with a scenario where the candidate nonsense mutation

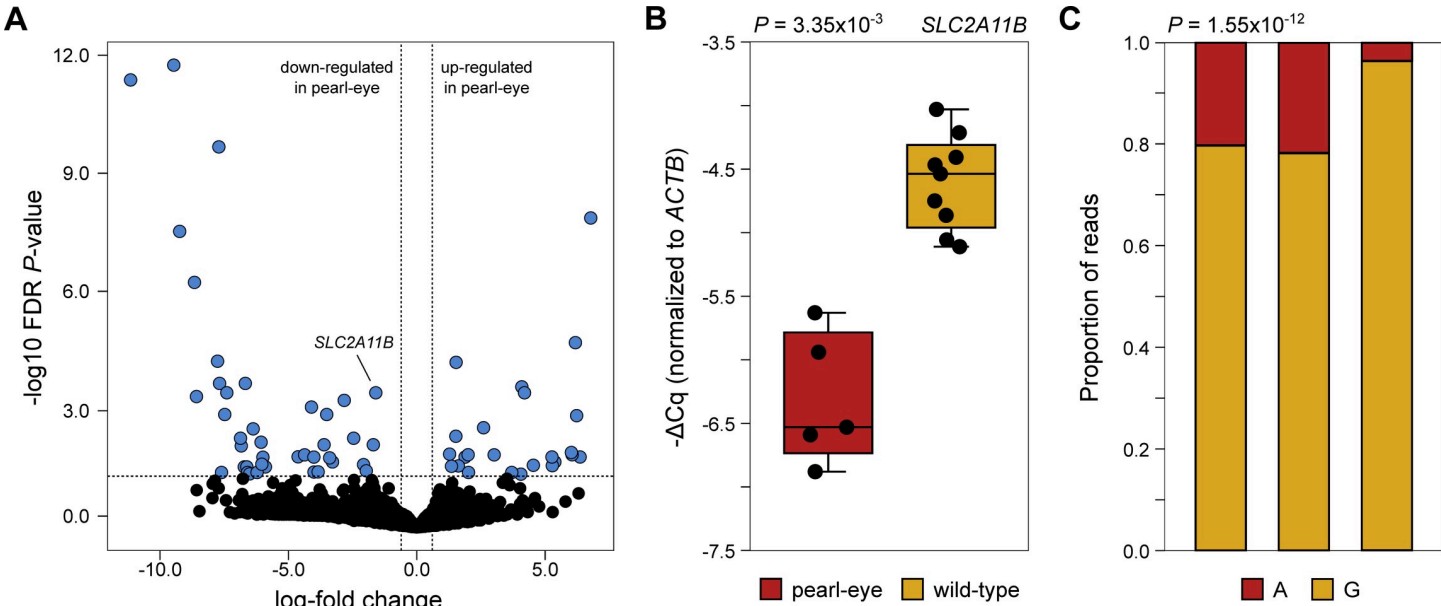

**Fig 4. Gene expression in wild-type and pearl-eye pigeons.** (A) Volcano plot with results from differential expression analysis of iris of the two phenotypes, comparing log-fold changes in expression with FDR-adjusted *P*-values. Significant results are highlighted in blue and *SLC2A11B* is indicated. (B) Differences in expression of *SLC2A11B* in the iris of both phenotypes as measured by RT-qPCR. Expression levels are normalized to the expression of beta-actin (*ACTB*). (C) Allele imbalance for the candidate causal mutation (AKCR02000030.1:1,895,934bp) in three pigeons heterozygous for the wild-type (G) and pearl-eye (A) alleles. The bars indicate for each individual the proportion of reads supporting each allele.

creates aberrant transcription products that might trigger nonsense-mediated mRNA decay. We note, however, that we cannot fully discard additional mutations affecting the expression of *SLC2A11B* contained within the pearl-eye haplotype.

## Assessing expression differences in genes associated with pterin metabolism and chromatophore differentiation

We finally used our RNA-seq dataset to investigate differences in gene expression in a (non-exhaustive) set of genes previously implicated or likely to interact with either pterin metabolism [20,21] or chromatophore differentiation [22–25]. None of the 21 genes considered were significantly up or down-regulated in our differential expression analysis, although some suggestive patterns were observed. For pterin metabolism genes, we found an apparent trend for expression differences in *GCHFR*, *SPR*, *AKR1A1*, *DHFR*, *PCBD1* and *QDPR* (S3A Fig, S4 Table). Nearly all these genes showed a trend for up-regulation in wild-type iris, with the exception of *GCHFR*, which is a negative regulator of *GCH1* (which in turn codes for the rate-limiting enzyme of the pterin synthesis pathway). In genes linked to chromatophore differentiation, minor differences were found for *PAX3*, *PAX7*, *SOX5*, *MITF* and *CSF1R*, all exhibiting a slight trend of up-regulation in wild-type iris (S3B Fig, S4 Table). Overall, our data suggests that there are no substantial changes in overall gene expression profiles between both iris types as well as in genes directly associated with pterin metabolism and/or chromatophore differentiation. This does not preclude the possibility that subtle changes in expression profiles occur (these could be more easily detected by increasing sample sizes and/or by investigating cell-specific patterns of expression).

## Discussion

The impressive variation in eye color of birds is the result of interactions between pigments (mainly melanins, carotenoids and pterins) and crystalline elements in the iris. The ultrastructure of the pigment cells of this tissue resembles the chromatophores of ectothermic vertebrates, but it is yet unclear if, and how, these different types of pigment cells in evolutionary distant lineages are related. By taking advantage of variation in eye coloration in domestic pigeons, we were able to map differences in iris pigmentation to an 8.5 kb region containing the gene *SLC2A11B*. A nonsense mutation leading to a premature STOP codon that segregates perfectly with eye color is the most likely causal mutation, and the observed differential expression of *SLC2A11B* between iris-types is likely the consequence of nonsense-mediated mRNA decay in pearl-eye pigeons. These results are confirmed in an independent study by [26] and contribute towards an increased understanding of the biology of pigmentation in vertebrates.

*SLC2A11B* is an excellent candidate gene to underlie pigmentation differences in the eyes of birds. This gene can be found in the genomes of birds and ectothermic vertebrates, but it has been lost in mammals [19]. Its product is a transmembrane protein that belongs to the GLUT family of facilitator transporters, which promote the transport of sugars and other carbon compounds across the cell membrane [27]. Its paralogue, *SLC2A11*, encodes a glucose and fructose transporter [28]. Although the precise cellular functions of *SLC2A11B* are poorly understood, another GLUT family member (*SLC2A9*) is known to mediate transport of urate, which is chemically similar to pterins. *SLC2A11B* might therefore act as a facilitator of pterin uptake in vertebrate cells. Importantly, *SLC2A11B* was also implicated in skin pigmentation in medaka fish with a putative role in xanthophore differentiation [19]. For example, results in medaka suggest a role upstream in the developmental path of pigment cells, since loss-of-function mutants exhibit alterations in xanthophores and other chromatophores. However, our differential expression analyses in the iris suggest that the overall transcriptional profiles of

wild-type and pearl-eye pigeons show only modest differences in the number of differentially expressed genes, and previous ultrastructural observations of both iris types revealed no significant differences in cell composition apart from the presence of yellow pigmented and non-pigmented cells [15]. Collectively, these results suggest that *SLC2A11B* in pigeons might exert its effects more proximate to pigment production or accumulation in xanthophores, rather than affecting a wide range of gene activities and other pigment cell types. Additional functional studies are, however, required to fully elucidate the role of *SLC2A11B* in pigmentation.

The essential role that *SLC2A11B* plays in coloration associated with xanthophore-like pigment cells in both pigeons and medaka is noteworthy. Despite many similarities, the pigment cells in the iris of pigeons and other birds also exhibit differences when compared to dermal chromatophores found in ectotherms, as they combine properties of both traditional xanthophores (pterin pigments) and iridophores/leucophores (guanine reflectors) [15]. Thus, the finding that the same gene that regulates differentiation of pterin and guanine-containing chromatophores in fish is also required for pterin-based pigmentation in the iris of pigeons raises the intriguing possibility that the molecular mechanisms generating coloration in the avian iris and in the integument of ectothermic vertebrates could often be shared. We therefore propose that the extensive catalog of genes that have been already implicated in pigmentation in fishes [19,29–31] can provide a treasure trove of promising candidate genes mediating the diversity of iris coloration existent in bird species in nature.

## Materials and methods

### Ethics statement

All experimental procedures were conducted in accordance with the Directive 2010/63/EU on the protection of animals. Animal care of birds kept in our facilities until dissection complied with national and international regulations for the maintenance of live birds in captivity (FELASA, Federation of European Laboratory Animal Science Associations). Birds were kept with *ad libitum* access to water and food. All protocols used were examined by the *Órgão Responsável pelo Bem-Estar Animal* (ORBEA) of CIBIO/InBIO.

### Whole-genome sequencing

To investigate the genetic basis of iris coloration in pigeons, we performed individual whole-genome sequencing of wild-type and pearl-eye birds. We sampled blood of unrelated pigeons from the wild-type (n = 14) and pearl-eye (n = 15) phenotypes from private pigeon breeders in Portugal (S1 Table). The blood samples were obtained by brachial venipuncture using a sterile needle into heparin-free capillary tubes, and subsequently stored in 96% ethanol until DNA extraction. Genomic DNA was extracted using an EasySpin Genomic DNA Tissue Kit SP-DT-250 (Citomed, Portugal). Prior to isolation, and after tissue lysis and digestion, RNA was removed with a RNAse A digestion step. DNA concentration of each extraction was quantified using both a NanoDrop instrument and a Qubit dsDNA BR Assay Kit.

Individual whole-genome sequencing libraries were constructed using published protocols [32,33]. We started by fragmenting DNA using a Covaris M220 with microTUBE-50 AFA Fiber Screw-Cap. Before library preparation, samples were normalized to have uniform DNA *concentrations across all samples*. Prior to the indexing of libraries, all libraries were analyzed with quantitative polymerase chain reaction (qPCR) to estimate optimal cycle settings on a Mx3005P qPCR System (Agilent Technologies). qPCR reactions were performed in 20µl reaction volume, containing 1 µl of 1:20 diluted DNA template, 0.1 U AmpliTaq Gold Polymerase, 1x PCR Buffer II and 2.5 mM MgCl$_2$ (all from Applied Biosystems), 0.5 mg/ml bovine serum albumin (Bio Labs), 0.2 mM dNTP Mix (Invitrogen), 0.2 µM each of the 5' nucleotide tagged Zeale forward and

reverse primers, and 1 μl of SYBR Green/ROX solution (Invitrogen). qPCR amplifications were performed on a Mx3005 qPCR machine (Agilent Technologies) with the following cycling conditions: 95°C for 10 min, followed by 40 cycles of 95°C for 30 s, 60°C for 60 s, and 72°C for 60 s.

Library indexing was carried out in 80 μl reaction volume, consisting of 8 μl of undiluted DNA template, 5 U of AmpliTaq Gold Polymerase, 1X PCR Buffer II and 2.5 mM of MgCl2 (all from Applied Biosystems), 0.2 mg/ml of bovine serum albumin (Bio Labs), 0.2 mM of dNTP Mix (Invitrogen), and 0.2 μM each of the forward and reverse BGI primers. Amplification was carried out with the following cycling conditions: 95°C for 12 min, followed by 14 cycles of 95°C for 20 s, 60°C for 60 s, and 72°C for 60 s, and a final extension of 5 minutes at 72°C [33]. We sequenced 5 lanes using 150 bp paired-end reads on a MGISEQ2000 at BGI Europe (Copenhagen). Whole-genome sequencing data are available in the Sequence Read Archive (www.ncbi.nlm.nih.gov/sra) under BioProject PRJNA679326.

For subsequent analysis, we also used publicly available whole-genome sequencing data from a previous publication [17], using breeds in which eye color phenotypes are fixed (S1 Table). These data were downloaded from the SRA repository (SRA054391) and treated the same way as our own sequencing data, as described below.

### Read mapping and variant calling

Sequencing reads were mapped to the rock pigeon reference genome assembly (Cliv_2.1; [18] with *BWA-MEM* [34] and default settings, followed by duplicate removal using *PICARD* (http://broadinstitute.github.io/picard). Variant calling was performed by means of the Bayesian haplotype-based method implemented in *Freebayes* v1.3.1 (https://github.com/ekg/freebayes). We modified the following additional parameters relative to the default settings: minimum mapping quality of 30, a minimum base quality of 30, turned off the left-alignment of indels, required at least three observations supporting the alternative allele, and required output of genotype qualities. We identified a total of 20,060,908 variants, including single nucleotide polymorphisms (SNPs), indels, multi-nucleotide polymorphisms, and complex polymorphisms. All variants were annotated using the genetic variant annotation and effect prediction toolbox *SnpEff* [35].

### Association mapping

For the genome-wide association analysis, we filtered the raw variant dataset extensively. First, we retained only SNPs. Second, only variants with quality scores of 500 or greater were retained. Third, we required a minimum coverage of 4 reads per individual and a maximum of 90 reads (i.e., three times the average coverage of the individual with higher coverage), otherwise a given genotype would be coded as missing. Fourth, all genotypes with genotype quality < 20 were coded as missing. Fifth, all variants with more than 20% missing data were removed. Finally, we excluded all variants with a minor allele frequency (MAF) > 0.1. All these filters were applied in this specific order using *VCFtools* [36] and resulted in a total of 4,411,065 SNPs for further analysis. Prior to performing the association analysis, we carried out genotype imputation using *BEAGLE* v5.1 with default parameters [37]. The association analysis was performed using a Fisher's exact test in $2 \times 2$ contingency tables using a recessive model implemented in the *R* package *SNPassoc* [38]. Bonferroni-corrected critical values were used for significance (0.05 / n, where n = 4,411,065; $P > 1.13 \times 10^{-8}$).

### Population genetics summary statistics

Genetic differentiation ($F_{ST}$) and nucleotide diversity (p) were estimated across the genome using a sliding window approach. These statistics were calculated using genotype probability

methods as implemented in *ANGSD* v0.929 [39]. We required reads with a mapping quality ≥ 20, an individual base quality ≥ 20, and a single mapping hit. We required a SNP *P*-value of $1\times10^{-6}$. The genome-wide results of both statistics presented in the "Results" section were derived from values averaged over 20 kb windows with a 4 kb step across each scaffold. Windows with less than 50% of the positions passing filters were excluded. A range of additional window sizes were used (5kb, 50kb, 100kb, and 200kb), but the results remained qualitatively unchanged with the top candidate region consistently emerging as the top outlier region.

### Genotyping

We performed Sanger sequencing of two SNPs on an Applied Biosystems 3130XL Sequencer following PCR amplification. First, to confirm whether the candidate region segregated perfectly with iris pigmentation in parent-offspring trios, we genotyped a SNP located within the IBD region (scaffold AKCR02000030.1:1,896,042) with the following primers: forward-AGTGCTATGCTGTAGGGCTA; reverse-CCTAAGGTACATTTTCTCCC. We genotyped 26 parent-offspring trios obtained from breeders in Portugal for a total of 78 samples. Second, we amplified a 409bp containing the candidate nonsense variant (AKCR02000030.1:1,895,934) using the following primers: forward-TTGGTTTTCAGGATTGAGGTG; reverse-AACCAC ATTGGAACAAACTGC). We genotyped a total of 92 individuals (40 pearl-eye, 52 wild-type), which includes the racing pigeon individuals used for whole-genome sequencing (see above). Blood sampling and DNA extraction were carried out as described above in the whole genome sequencing section.

### RNA extraction and transcriptome sequencing

To study gene expression in the iris of pearl-eye and wild-type pigeons, we sampled 14 animals (5 pearl-eye and 9 wild-type). Birds were anaesthetized using isoflurane inhalation and euthanized by manual cervical dislocation, following guidelines by the AVMSA (American Veterinary Medical Association). Irises were dissected and snap frozen in liquid nitrogen. Tissues were stored at -80˚C until RNA extraction. Total RNA was isolated using the RNeasy Mini kit (Qiagen Sciences Inc, Germantown, MD) followed by DNAse digestion. RNA integrity was measured using a TapeStation RNA ScreenTape (Agilent) and RNA concentration was calculated using the Qubit RNA BR assay kit. cDNA was generated from approximately 1 μg of RNA with the GRS cDNA Synthesis Kit (GRiSP, Porto, Portugal), according to the manufacturer's instructions.

A subset of the birds sampled for RNA extraction (two pearl-eye, three heterozygous wild-type and one homozygous wild-type) were used for RNA sequencing. From each individual strand-specific Illumina libraries were prepared using the LM-seq method [40] and sequenced using 150 bp paired-end reads. Sequencing quality was checked using *FastQC* v0.11.8. Reads were corrected with *Rcorrector* v1.0.3.1 [41] to exclude read pairs with at least one unfixable read, and *Trim Galore*! v0.6.0 (https://www.bioinformatics.babraham.ac.uk/projects/trim_galore) to remove adapters, low quality bases (Phred score <5), and reads smaller than 36 bp. We generated a total of 558,644,610 reads with an average of 93,107,435 reads per individual (S2 Table). RNA-seq data are available in the Sequence Read Archive (www.ncbi.nlm.nih.gov/sra) under BioProject PRJNA679326.

### *de novo* transcriptome assembly

The pigeon reference assembly and annotation were derived from a pearl-eye individual, and are thus based on *SLC2A11B* transcripts harboring our candidate nonsense mutation. To investigate possible differences in isoforms between pearl-eye and wild-type pigeons, we

assembled transcriptomes for one homozygous bird of each phenotype. For the pearl-eye phenotype, we selected the individual with the highest number of reads. Before assembly, reads were mapped with *Bowtie2* v2.3.5 [42] to the SSUParc and LSUParc fasta files from the database SILVA [43] (https://www.arb-silva.de/; downloaded April 2019) using the very-sensitive-local option. Reads that were positive hits were discarded from our dataset to remove contamination with ribosomal RNA. For assembly, we used *Trinity* v2.8.4 [44] with kmer size 35, 45, 55, 65, 75 and 85, using only paired-end reads. Of the resulting transcripts, we excluded those that were smaller than 1,000 bp. To identify *SLC2A11B* transcripts, we mapped the remaining transcripts to the reference genome with *HISAT2* v2.2.1 [45] and retained only those that mapped to the expected genomic region on scaffold AKCR02000030.1. The relative abundance of each transcript (transcripts per million, TPM) in the original RNA-seq reads of each of the two individuals was estimated using the quasi-mapping approach implemented in *Salmon* v1.2.1 [46], with the options *seqBias* and *useVBOpt*. For each individual, we retained only transcripts that represented more than 10% of the overall abundance. Resulting transcripts were manually aligned using *BioEdit* v7.2.5 [47]. We complemented the transcriptome assembly by investigating differences in splice junction arrangements in each of the two birds. For this we mapped the corrected and trimmed reads to the reference rock pigeon genome with *HISAT2* v2.2.1. We then extracted reads contained within the interval AKCR02000030.1:1,892,000–1,904,000 and constructed a sashimi plot using *ggsashimi* [48].

## Analysis of differential expression using RNA-seq

We used RNA-seq reads from two pearl-eye and four wild-type pigeons to quantify overall gene expression patterns in the iris. From each individual, we estimated the relative abundance of each transcript (TPM) from the published reference transcriptome using *Salmon* v1.2.1, with the options *seqBias* and *useVBOpt*. Calculation of differential gene expression was performed with *edgeR*, through *DEApp* [49]. We excluded transcripts with <1 TPM per million in at least two samples. We used a false discovery rate (FDR) of 5% and considered only transcripts with log-fold change over 1.5.

## Expression levels of *SLC2A11B* using RT-qPCR

We confirmed differences in relative expression levels of *SLC2A11B* through reverse transcription quantitative polymerase chain reaction (RT-qPCR). Primers were designed to amplify a 194-bp portion of the coding region, spanning across exon-exon boundaries to prevent contamination from genomic DNA (forward: AATTCAGGTGTTGGGCTCTG; reverse: GGGA AACAGCTGCTGGATAA). To standardize expression levels, we amplified the house-keeping gene beta-actin (*ACTB*, RT-qPCR primers from [50]). We performed three technical replicates for each individual and for both genes using the iTaq Universal SYBR Green Supermix (Bio-Rad laboratories) and an CFX96 Touch Real-Time PCR Detection System (Bio-Rad laboratories). Quantification cycle (Cq) values of the replicates were averaged, and Cq values of *SLC2A11B* were standardized to the expression of beta-actin using a -ΔCq approach [51]. We tested for significant differences between pearl-eye and wild-type expression of *SLC2A11B* with a Mann-Whitney U test (differences were considered significant if $P < 0.05$).

## Allelic imbalance

We investigated allele-specific differences in the expression of *SLC2A11B* between haplotypes harboring the pearl-eye and wild-type variants using the RNA-seq data. We mapped reads from the three heterozygous birds to the reference assembly using the splice-aware aligner *HISAT2* v2.2.1, and retrieved counts of reads supporting the pearl-eye and wild-type alleles at

the candidate causal mutation for each individual. We tested for a significant ($P < 0.05$) deviation from an equal proportion of alleles (null hypothesis) using a Fisher's exact test.

## Supporting information

**S1 Fig. Alignment of multiple bird genomes in the candidate nonsense mutation region.** The candidate mutation is highlighted in orange.
(PDF)

**S2 Fig. Exon structure of *SLC2A11B* from one wild-type and one pearl-eye homozygous pigeons that were used for transcriptome assembly. (A)** Alignment of assembled transcripts of *SLC2A11B* to zebra finch and chicken transcripts from the *Ensembl* database (retrieved on November 2020). The putative canonical transcript of *SLC2A11B* in pigeons, and its respective predicted open reading frame, is shown on top. Of all the transcripts that were assembled, we only considered those that represented more than 10% of the overall transcripts mapping to the candidate region. Colors indicate the base content of each sequence: adenine–green; guanine–black; cytosine–blue; thymine–red. **(B)** Sashimi plots based on reads mapping to the genomic interval containing *SLC2A11B* for the two pigeons. Compared to the wild-type, the pearl-eye individual has a relatively decreased amount of reads in the 5' end of the gene, different proportions of reads along the transcript and relatively increased amount of reads in intronic regions.
(PDF)

**S3 Fig.** Gene expression profiles between pearl-eye (red) and wild-type (yellow) pigeons for candidate genes for skin color development in ectothermic vertebrates. (A) Relative expression levels (TPM, transcripts per million) of genes in the pterin synthesis pathway. (B) Similar to panel (A) for a group of genes previously implicated in xanthophore differentiation in ectothermic vertebrates.
(PDF)

**S1 Table. Sequencing summary statistics of samples used for whole-genome analyses.**
(PDF)

**S2 Table. Summary statistics of samples used for RNA sequencing and allele counts at the candidate causal mutation (AKCR02000030.1:1,895,934bp).** Genotypes were determined through Sanger sequencing.
(PDF)

**S3 Table. List of differentially expressed genes between the iris of wild-type and pearl-eye pigeons.** Negative fold-change values indicate down-regulation in pearl-eye iris. Linkage group information follows the genetic map of [18]. Genes located in linkage group 20 (the same as the putative causal locus) are marked in **bold**.
(PDF)

**S4 Table. Individual expression values (TPM, transcripts per million) for a set of genes implicated in pterin metabolism and chromatophore differentiation (see also S3 Fig).** TPM values were calculated using the quasi-mapping approach implemented in the software *Salmon* [46].
(PDF)

## Acknowledgments

We thank pigeon breeders who provided samples for this study, namely João Paulo Valente, Miguel Mendonça, Marta Lourenço, Mário Soares, Pedro Porto, Rui Cruz and *Centro*

*Internacional de Criação de Pombos.* We thank *Ibanidis Lda* (*Versele-Laga* distributor) for food supplies. M.A.G. thanks Shyam Gopalakrishnan for all computational support during her visit in Copenghagen. Paulo Pereira also helped with the analysis of transcriptomic data.

## Author Contributions

**Conceptualization:** Pedro Andrade, Małgorzata A. Gazda, M. Thomas P. Gilbert., Miguel Carneiro.

**Data curation:** Pedro Andrade, Małgorzata A. Gazda, Pedro M. Araújo, Cristiana I. Marques, Miguel Carneiro.

**Formal analysis:** Pedro Andrade, Małgorzata A. Gazda, Cristiana I. Marques, Miguel Carneiro.

**Funding acquisition:** Małgorzata A. Gazda, Cristiana I. Marques, Ricardo J. Lopes, M. Thomas P. Gilbert., Miguel Carneiro.

**Investigation:** Pedro Andrade, Małgorzata A. Gazda, Pedro M. Araújo, Sandra Afonso, Jacob. A. Rasmussen, Cristiana I. Marques, Ricardo J. Lopes, M. Thomas P. Gilbert., Miguel Carneiro.

**Resources:** Pedro M. Araújo, M. Thomas P. Gilbert., Miguel Carneiro.

**Supervision:** M. Thomas P. Gilbert., Miguel Carneiro.

**Visualization:** Pedro Andrade, Pedro M. Araújo, Miguel Carneiro.

**Writing – original draft:** Pedro Andrade, Małgorzata A. Gazda, Miguel Carneiro.

**Writing – review & editing:** Pedro Andrade, Małgorzata A. Gazda, Pedro M. Araújo, Sandra Afonso, Jacob. A. Rasmussen, Cristiana I. Marques, Ricardo J. Lopes, M. Thomas P. Gilbert., Miguel Carneiro.

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
