## [Decision Letter · Decision Letter 0]

6 Jan 2021

Dear Dr Andrade,

Thank you very much for submitting your Research Article entitled 'Molecular parallelisms between pigmentation in the avian iris and the integument of ectothermic vertebrates' to PLOS Genetics.

The manuscript was fully evaluated at the editorial level and by independent peer reviewers.  This is a very well written paper with clear findings that will be of broad interest to the genetics community.  The suggestions made by the reviewers largely concern the presentation of the results in the figures, and should be straightforward to address.  We therefore ask you to modify the manuscript according to the review recommendations. Your revisions should address the specific points made by each reviewer.

[LINK]

Yours sincerely,

Kelly A. Dyer

Associate Editor

PLOS Genetics

Bret Payseur

Section Editor: Evolution

PLOS Genetics

Reviewer's Responses to Questions

**Comments to the Authors:**

Reviewer #1: This is a really nice gene study that identifies an interesting gene required for normal pigmentation of the pigeon iris. The avian iris is of considerable interest in the field of pigmentation research as it is thought to harbor evolutionarily ancient chromatophore classes found in ectothermic vertebrates but not elsewhere in endotherms. So any study identifying genes required will iris color makes a substantive contribution to the field. In this particular case, the authors have convincingly shown that SLC2A11B has a premature stop codon in pearl mutant pigeons, which lack yellow pteridines. This particular gene is notable for having been identified several years ago as essential to yellow coloration of xanthophores and leucophores in medaka fish, and so the analyses of Andrade et al. really set the stage for future investigations into conservation of cell lineage and differentiation pathways across a deep phylogenetic divide.

The authors have done a thorough job in their gene identification, using a combination of whole-genome resequencing, mapping by several different metrics, RNA-seq and targeted assessment of downstream gene expression. The study is rigorous and the manuscript is exceptionally clear, making it a real pleasure to read. The only small reservation I had concerned the presentation of candidate pteridine and chromatophore associated gene expression in Figure 4D,E, in which there were some trends but not significant differences. Since these data are not normalized by cell type, but instead represent a complex mixture of cells in the iris, and sample sizes are very small, they really do not contribute substantially. So I would be more inclined to move these panels to a supplementary figure, with the appropriate caveats so they are not over-interpreted by readers.

Although I am very enthusiastic about the paper, I can imagine one might object that the authors have not truly demonstrated a "mechanism" by which SLC2A11B promotes pteridine accumulation, or normal development of the relevant chromatophores. Indeed, it will be exciting to learn these sorts of details. Realistically, however, it seems extremely unlikely that the kinds of analyses that would be required to extract such information could be accomplished in a reasonable time-frame, as they would likely require the authors to find new collaborators or develop new techniques, as well as additional experimental material and perhaps even another model like the original medaka mutant, or a corresponding mutant in zebrafish, either of which would be more amenable to such next steps than pigeon. These would not be trivial undertakings in normal times but are likely to be especially difficult during a pandemic.

Reviewer #2: This is a very convincing and exciting study of pigmentation genomics in pigeon iris coloration. The results were very clear and drawn from a range of data types. I found that the interpretation was fair and, while some could take issue with the sample sizes for the RNA studies, this is logistically challenging and, moreover, are more of supporting data to the clear genomic results. Finally, the paper is exceptionally well-written and makes for a very compelling manuscript—I do not have any specific suggestions for improvement.

My only suggestion would be to add somewhere (likely figure 3?) the intron-exon structure of the SLC2A11B (and any other nearby genes). Basically the results refer to SLC2A11B as being the only gene in this region, however the reader never really gets to 'see' this result directly, as well as any other nearby genes. Thus, including this annotation information visually would be an improvement.

Reviewer #3: Review is uploaded as an attachment

**Have all data underlying the figures and results presented in the manuscript been provided?**

Reviewer #1: Yes

Reviewer #2: Yes

Reviewer #3: Yes

PLOS authors have the option to publish the peer review history of their article (what does this mean?). If published, this will include your full peer review and any attached files.

Reviewer #1: No

Reviewer #2: No

Reviewer #3: No

---

## [Decision Letter · Decision Letter 1]

29 Jan 2021

Dear Dr Andrade,

Thank you very much for submitting your Research Article entitled 'Molecular parallelisms between pigmentation in the avian iris and the integument of ectothermic vertebrates' to PLOS Genetics.

The manuscript was fully evaluated at the editorial level and by one of the previous reviewers. We appreciate your attention to the reviewer's suggestions, and we expect that once you address the one remaining concern about identifying the genetic linkage group for each one of your differential expression hits your article will be accepted. Please see the comments from the Reviewer and their analysis (attached) that addresses this point. Given this is a genetic mapping paper, we agree with the reviewer that this analysis would strengthen the paper.

We therefore ask you to modify the manuscript according to the review recommendations. Your revisions should address the specific points made by each reviewer.

[LINK]

Yours sincerely,

Kelly A. Dyer

Associate Editor

PLOS Genetics

Bret Payseur

Section Editor: Evolution

PLOS Genetics

Reviewer's Responses to Questions

**Comments to the Authors:**

Reviewer #3: The authors have responded to main point #1 by highlighting the differential expression hits in Supplementary Table 3 that are on the same scaffold as SLC2A11B. This is a step in the right direction. However, this is not what I had originally suggested: I had suggested the authors determine which genetic linkage group each hit belongs to, not which scaffold (indeed the latter is already clear from the table). This is because linkage groups typically span a whole chromosome whereas scaffolds do not. Granted, the scaffold the causative haplotype resides on is a large one (8.6 Mb), and the haplotype is about 1.8 Mb from the scaffold end (so it does seem likely that any putative cis-regulated gene would be located on the scaffold). However, given that A) enhancers can sometimes act at megabase distances, and B) scaffolds are sometimes misassembled, it does not seem unreasonable to be curious whether any of the differential expression hits reside on different scaffolds, which are linked to the one in question. To illustrate what I mean, I have attached the results of a simple analysis, where I downloaded Table S6 from the Holt et al paper, which contains a list of scaffolds and their corresponding linkage groups, and did a simple data join in R to determine the linkage group for each of the genes in Supplementary Table 3. SLC2A11B is on linkage group 20 and indeed the only other hits on that linkage group are on the same scaffold (CHEK2 and TRAFD1). This corresponds to what the authors noted in their response to Main Point #1 (but note the improvement in that we can now make a statement about the entire linkage group). Of the 64 genes, the linkage group for all but 8 could be determined from the Holt et al results. This seems to me to be worth including in the paper. Given that it is such an easy thing to include, I cannot recommend publication until the authors either include it or explain why it should not be included.

Aside from that, I have no further issues to bring up. The matter of the title I will leave to the authors and editors. This was a fun paper to read. I think if the linkage group data is included it will be even better.

**Have all data underlying the figures and results presented in the manuscript been provided?**

Reviewer #3: Yes

PLOS authors have the option to publish the peer review history of their article (what does this mean?). If published, this will include your full peer review and any attached files.

Reviewer #3: No

---

## [Editor Report · Decision Letter 2]

8 Feb 2021

Dear Dr Andrade,

We are pleased to inform you that your manuscript entitled "Molecular parallelisms between pigmentation in the avian iris and the integument of ectothermic vertebrates" has been editorially accepted for publication in PLOS Genetics. Congratulations!

Yours sincerely,

Kelly A. Dyer

Associate Editor

PLOS Genetics

Bret Payseur

Section Editor: Evolution

PLOS Genetics

Comments from the reviewers (if applicable):

**Data Deposition**

http://datadryad.org/submit?journalID=pgenetics&manu=PGENETICS-D-20-01786R2

**Press Queries**

---

## [Editor Report · Acceptance letter]

18 Feb 2021

PGENETICS-D-20-01786R2 

Molecular parallelisms between pigmentation in the avian iris and the integument of ectothermic vertebrates 

Dear Dr Andrade, 

We are pleased to inform you that your manuscript entitled "Molecular parallelisms between pigmentation in the avian iris and the integument of ectothermic vertebrates" has been formally accepted for publication in PLOS Genetics! Your manuscript is now with our production department and you will be notified of the publication date in due course.

With kind regards,

Alice Ellingham

PLOS Genetics

On behalf of:
